# Density-equalized RANDT scan matching with integrated outlier removal and point density uniformity

**Anila Johnson[1], Umashankar Subramaniam[2], HyungSeok Kim[3], Divya Udayan J.**[1]*

**1** Department of Computer Science and Engineering, Amrita School of Computing, Amrita Vishwa Vidyapeetham, Amritapuri, India, **2** College of Engineering, Prince Sultan University, Riyadh, Saudi Arabia, **3** Konkuk University, Seoul, South Korea

\* divyaudayanj@am.amrita.edu

## Abstract

The significance of three-dimensional point cloud data in generating high precision point cloud maps for environmental sensing, geo-spatial analysis and autonomous driving is increasing today with the advancements in sensing technologies such as LiDAR. Accurate registration is required to reduce noise and maintain uniform point density. The existing algorithms for this purpose shows limitations such as slow convergence, partial overlaps, and convergence to local minima or maxima as a consequence of the non-homogeneous distribution of feature points in the point cloud scenes. To tackle these problems we present a novel scan matching framework Robust and Adaptive Normal Distribution Transform (RANDT). In our method an incremental scan matching module is introduced which continuously perform scan matching with newly matched scans to achieve uniform and dense point distribution across the scene. Prior to the scan matching process an outlier removal feature which removes the noisy data points is also included to achieve more accurate point cloud data. The evaluations on KITTY and ModelNet40 datasets demonstrate that proposed RANDT method achieves RMSE values of 0.054 m and 0.062 m with an error reduction of 18–25% even with noise and partial overlaps, and also attains highest point density uniformity score 0.92 than other baseline methods.

## Introduction

High-definition 3D mapping forms the foundation for numerous contemporary technologies, including autonomous navigation, augmented reality, robotic exploration, geo-spatial analysis and smart city systems [1,2]. These applications demand precise environmental representation and accurate localization, making reliable and accurate 3D mapping and registration methods essential requirements. Autonomous systems may already function well in complicated, unstructured situations thanks to the widespread availability of affordable 3D sensors like LiDAR [3]. Due to its capacity

**Data availability statement:** The data used in this study are publicly available and can be accessed at: https://github.com/anilabibin/Point_Cloud_registration.

**Funding:** The author(s) received no specific funding for this work.

**Competing interests:** The authors have declared that no competing interests exist.

to immediately record precise geometric features of the surroundings, LiDAR has become a popular option for 3D mapping among the widely used sensing modalities, including cameras, radar, and LiDAR [4].

LiDAR offers high-resolution, dense, and geometrically consistent data, whereas radar systems are frequently susceptible to noise and interference, and camera-based systems depend on computationally demanding methods to recreate 3D geometry from 2D images [5, 6]. But no sensor is perfect. To create logical and high-fidelity maps, all obtained 3D scans need to be post-processed and aligned to a common coordinate system, especially for long-term operations and real-time robotic applications [7]. Robots can carry out mission-critical activities including mobile manipulation, obstacle avoidance, cave and underwater exploration, and search and rescue operations in disaster areas thanks to these intricate 3D maps. Simultaneous Localization and Mapping (SLAM) systems, which localize an unknown space while localizing the agent's location within it, often incorporate these capabilities [8,9].

Point cloud registration or scan matching aims to transform the collected LiDAR scans to a common coordinate system. A point cloud is a collection of three-dimensional points in Cartesian space. It depicts the three dimensional surface exposure of an object or environment. Effective point cloud registration is essential for many applications such as motion planning for autonomous navigation,creating point cloud maps for geo-spatial analysis. Most of the recent studies [10,11] have concentrated on increasing the speed of correspondence, eliminating partial overlaps, eliminating noisy points, and other related issues. Existing point cloud registration algorithms tackle these issues and improve positioning accuracy by aligning newly captured scans with existing data or reference maps [12].

Among the available techniques, the Iterative Closest Point (ICP) algorithm is commonly used because of its simplicity of usage [13]. However, ICP performance degrade when confronted with substantial initial pose mismatches, limited scan overlap, or fault measurements. Probabilistic methods such as Coherent Point Drift (CPD) [14] and Normal Distributions Transform (NDT) [15] have pop up as alternatives scan matching methods to improve the existing methods and overcome these challenges. By statistically modeling the spatial distribution of point clouds, these methods deliver high level robustness and convergence characteristics by statistically modeling the spatial distribution of point clouds.

While many algorithms are developed for point cloud registration, no existing studies on point cloud registration has focused on the problem of point density uniformity along with outlier removal which is essential for high definition three dimensional point cloud maps [16–18]. This indicates the need for more accurate registration methods. In practical applications, factors such as movement of laser scanner, scan strips overlap, altitude variations and weather conditions create uneven point distributions. This may degrade the overall accuracy of the scene and negatively affects subsequent tasks like object recognition, feature extraction, autonomous navigation, forestry, vegetation analysis and so on [19,20]. The non-uniform distribution of point density in point cloud data has direct impact on many downstream tasks. Sparsely distributed region will lead to unreliable local geometric descriptors and unstable



surface normal estimation. This will lead to an increase in alignment error and reduce the stability of convergence in scan matching process. Conversely dense distributed regions typically lead to bias feature extraction and can dominate the optimization process which results in registration drift and reduced robustness in irregular environments [21,22]. Some recent works have also confirms that that non-uniform sampling significantly degrades 3D object detection and scene understanding performance. This is because learning-based models tend to overfit high-density areas while underperforming on sparsely sampled structures [23].

Our work introduce an improved outlier resilient NDT for uniform density registration RANDT (Robust and Adaptive Normal Distribution Transform), which is an NDT-based scan matching algorithm that explicitly prioritizes point density uniformity as a main objective. The RANDT incorporate probabilistic modeling of scan points, local normal distributions computation, transformation of new scans into a reference coordinate system, and outlier removal to ensure reliable correspondences.

RANDT provide enhanced scan matching system that is well suited for real-time mapping applications in autonomous systems, digital model creation and reconstruction and feature extraction. The approach accomplishes this along with preserving computational efficiency, making it suitable for time-critical applications such as autonomous driving, digital twin generation, and disaster response applications. The overall performance of the proposed RANDT technique is evaluated using two different indoor and outdoor LiDAR datasets collection and simulated environments. The findings reveal substantial improvements in both registration precision and point cloud density uniformity when compared to conventional methods. The main contributions of our work are:

- A variant of normal distribution based scan matching algorithm which is designed to serve as a reliable platform for high definition LiDAR map applications.

- We developed an Incremental scan matchhing framework for preserving uniform point density across the scans, which is essential for producing high-quality three-dimensional LiDAR maps.

- An outlier removal method which improves the accuracy of point cloud matching and precision of point correspondences during the scan matching process.

- An efficient algorithm that estimates the transformation parameters is provided for transformation of moving scan to the reference scan.

- The suggested RANDT method improves the accuracy of point cloud matching and precision of point correspondences compared to existing techniques.

- To the best of our knowledge, no prior work unifies these components within a single point-cloud registration pipeline.

The integration of Incremental scan matching and IQR(Inter Quartile Range) based outlier removal provides a novel unified framework for the point cloud registration process which improves both registration accuracy and robustness. No existing method integrates all of these components within a unified point-cloud registration process.

The rest of this paper is organized as follows: A review of related research on scan matching and registration methods is summarized in the next part. Following that a detailed description of the suggested methodology is provided. The performance analysis and experimental findings are discussed in the Result section. Last section concludes the paper and outlines potential directions for future research.

## Related works

Scan matching and point cloud registration have been long-standing challenges in computer vision, robotics, and 3D mapping. Many algorithms have been put forward to match partially overlapping point clouds with geometric consistency and computational efficiency over the years. This section surveys traditional methods, probabilistic methods,

optimization-based methods, and recent deep learning methods, focusing especially on techniques closely related to our research.

## Classical methods

The Iterative Closest Point (ICP) algorithm is arguably the most commonly known registration method, originally created to register two point clouds by refining correspondences iteratively and reducing the Euclidean distance between pairs [13]. Its nature and simplicity meant that it became ubiquitous. ICP is highly initialization-sensitive and tends to settle to local minima in cases of significant misalignments or noise [24,25]. Further, ICP does not treat outliers strongly and performs poorly in non-uniform density situations. Sub-variants like Generalized ICP and Piecewise ICP were developed to overcome these limitations [16].

Piecewise ICP extends the traditional ICP framework through a decomposition strategy that partitions the point set into smaller constituent regions prior to alignment. This approach yields improved robustness when operating in structured environments, offering better convergence characteristics and more precise local geometry representation than traditional ICP [26]. However, this enhancement comes at the cost of increased computational overhead that grows with partition count, while maintaining sensitivity to noise artifacts and non-uniform point density variations [27].

Enhancements on classical point cloud registration methods focuses on accelerated ICP variants like point to plane multi-resolution ICP and GPU-ICP [28]. It is best suitable for large scale LiDAR datasets since it improves convergence speed. To improve the robustness under partially overlapped or sparsely distributed scans, a hybrid geometric frameworks that integrate voxel hashing and adaptive correspondence filtering have also introduced [29].

## Feature based methods

Feature-based registration techniques are designed to address the weakness of purely geometric alignment by initially extracting descriptive features from the point cloud and then finding correspondences among these features [30,31]. Rather than comparing raw points directly, these techniques are based on invariant descriptors capturing the local or global scene structure. Spin Images, Point Signature, and the more popular Fast Point Feature Histogram (FPFH) are well-known descriptors [32]. After establishing correspondences, algorithms like RANSAC or Fast Global Registration (FGR) are utilized to estimate the transformation robustly [33,34].

Since features capture geometric context outside of raw point coordinates, feature-based registration's main strength is its ability to handle large misalignments and partial overlaps [35]. For instance, FPFH together with RANSAC has worked well in achieving coarse global alignment even when datasets are noisy. Furthermore, these approaches give a good starting point to fine local refinements such as ICP and hence are common in two-stage registration pipelines.

Some of the feature based registration approaches incorporate graph-structured descriptors, like GCN-based local embeddings and GeoDesc [36]. It can capture higher order geometric relationships which improves correspondence reliability. To achieve robust matching under viewpoint changes and significant point sparsity methods like 3D3 and YOHO employ orientation-aware feature learning and hierarchical neighbourhood aggregation [37].

However, feature-based approaches also have drawbacks. To begin with, descriptor reliability is highly sensitive to the quality of the input data. Noisy or sparse point clouds tend to produce weak or ambiguous features, particularly in the presence of repetitive structures or low texture. Moreover, feature extraction per se can be computationally costly, especially for high-resolution scans. Third, global alignment is achievable though the precision of the transformation tends to be less than fine-grained methods like NDT or ICP and needs to follow a refinement stage.

## Probabilistic and distribution-based methods

To overcome the limitations of ICP, probabilistic methods were implemented. Among them, the Normal Distributions Transform (NDT) emerged as a leading solution by mapping local point neighborhoods to Gaussian distributions to smooth

noise effects and provide faster convergence [15]. While NDT improved robustness, it also caused discretization-related errors and required careful tuning of grid parameters [38,39].

The 3D Multi-Normal Distribution Transform (3DMNDT) expanded this concept by extending distribution modeling into three dimensions using multi-modal representations, enabling improved precision in complex environments [17]. Its advantage lies in achieving an effective balance between robustness and accuracy while reducing dependence on initial alignment. 3DMNDT, however, needs substantial memory because it stores multiple distributions per voxel and its runtime remains non-trivial for big-scale scans, which constrains its application in real-time robotics or dense 3D mapping [40].

The advancements in NDT based scan registration including multi-resolution voxel distribution models and Gaussian mixture-regression–based registration improves robustness against non-uniform sampling [41,42]. Also the point cloud registration methods employing kernel adaptive weighting have shown improved accuracy in highly dynamic or cluttered urban environments.

## Robust estimation approaches

RANSAC-based methods were devised to address the issue of point correspondence outliers. The basic concept is that subsets of points are sampled iteratively, transform estimates are made, and they are tested against consensus sets. Classical RANSAC is robust but computationally costly, particularly with large data sets [33].

The One-Point RANSAC algorithm is highly efficient in limiting this computational cost by obtaining candidate transformations from minimal point correspondences, thus speeding up the consensus-building step [43]. This development allows for faster registration on outlier-prone data at the expense of reasonable accuracy. One-Point RANSAC loses robustness when point density is sparse or geometric features are less unique. The randomness involved in sampling still brings variability to performance and makes it less than ideal for high-accuracy applications [44].

In addition, robust kernels and probabilistic consistency filters such as Cauchy weighting [45], trimmed ICP, and correspondence rejection via geometric compatibility have shown strong resilience against structured outliers and partial overlaps [46].

## Deep learning-based registration

Deep learning has in recent years been used in point cloud registration to introduce direct learning methods for features and correspondences [47]. DeepSIR is one such method that uses a neural network to learn strong correspondences while tolerating noise and partial overlap [48,49]. Deep learning techniques exhibit strong generalization over multi-different datasets and are capable of surpassing conventional methods when adequate training data is provided [50].

Although successful, learning-based approaches like DeepSIR have significant drawbacks. They need large-scale labeled training sets, which do not exist in many real-world problems. Inference also tends to be slower compared to classical approaches as a result of neural architecture complexity, and memory usage is considerably larger [51,52]. Generalization is also an issue with performance dropping when seen data distributions are not encountered during training.

Recent deep learning methods including Predator, GeoTransformer, and CoFiNet incorporate attention driven mechanisms to extract robust correspondences under low-overlap conditions [53–55]. The introduction of diffusion-based estimators and GNN-based alignment frameworks further enhance registration reliability and dataset generalization [56].

Some other popular methods include Coherent Point Drift (CPD), which formulates registration as a probability density estimation task, registering points by representing one set as centroids of a Gaussian Mixture Model [14,57]. CPD contains smooth alignment, albeit with higher computational complexity, especially for non-rigid deformations. Feature-based approaches, such as Fast Point Feature Histograms (FPFH) using RANSAC, enable coarse registration by selecting local descriptors but are generally sensitive to noise and perform poorly in repetitive scenes. Hybrid approaches that combine coarse global registration with fine local refinement (e.g., FGR+ICP) attempt to utilize the strengths of a variety of methods but also inherit complexity and parameter tuning overheads.

The above discussions on existing methods underscores the need for a point cloud registration framework that simultaneously ensures outlier robustness, uniform point density, and high computational efficiency. The RANDT scan matching presented here overcomes the tried to overcome the existing issues of uneven point distribution and noise disturbances. By redistributing points uniformly and adapting robust noise-handling methods, RANDT achieves improved density preservation (minimal NDI), better runtime, and reduced memory usage compared to state-of-the-art approaches. In contrast to ICP or its derivatives, RANDT is not trapped by local minima; in contrast to RANSAC-based solutions, it is determinism; and in contrast to learning-based approaches, it is not dependent on large-scale training. RANDT thus provides a well-balanced, generalizable, and real-time-capable solution to 3D scan matching. Fig 1 illustrates the development of several scan matching algorithms and their salient characteristics from the conventional to the suggested method.

## Methodology

Three complementary modules are integrated into the proposed scan matching framework to achieve robust, noise-resilient, and dense point cloud registration: an incremental scan matching strategy for density equalization, an extended normal distributions transform with outlier removal and a singular value decomposition (SVD) based alignment refinement. Fig 2 depicts the whole architecture of the proposed method.

### Extended Normal Distribution based Scan Matching (ENDSM)

Sensor noise sometimes creates sparse outliers that decrease registration accuracy throughout the 3D mapping process. We add an outlier elimination mechanism based on the Interquartile Range (IQR) to the Normal Distributions Transform (NDT) framework in order to get around this restriction. For a stable NDT based scan matching outlier removal is necessary, especially when point density varies across the scene. Existing adaptive filters such as Statistical Outlier Removal (SOR) or radius-based methods are also efficient, but they depends on user-defined parameters and are sensitive to local density variations. However, the IQR threshold provides a simple, distribution-free, and robust statistic that suppresses extreme deviations without assuming Gaussian noise. So IQR based outlier removal is efficient and tuning-free alternative for handling non-uniformly distributed LiDAR point clouds. Fig 3 shows the impact of outlier removal in a sparse distributed point cloud.

Algorithm 1 introduces this extension, which is the Extented Normal Distribution Scan Matching(E-NDSM). The input reference scan is divided into uniform voxels in the first step. The set of points in a voxel is compactly represented by its mean vector and covariance matrix, creating a multivariate Gaussian distribution, rather than matching every point within each voxel. For each voxel, quartiles are computed over the scalar probability density values obtained from the multivariate Gaussian within each voxel, instead of computing voxels individually on x, y, and z coordinate dimensions. Hence it provides a single scalar metric for outlier evaluation. Since the pair $(\mu, \Sigma)$ can statistically describe the entire voxel, this

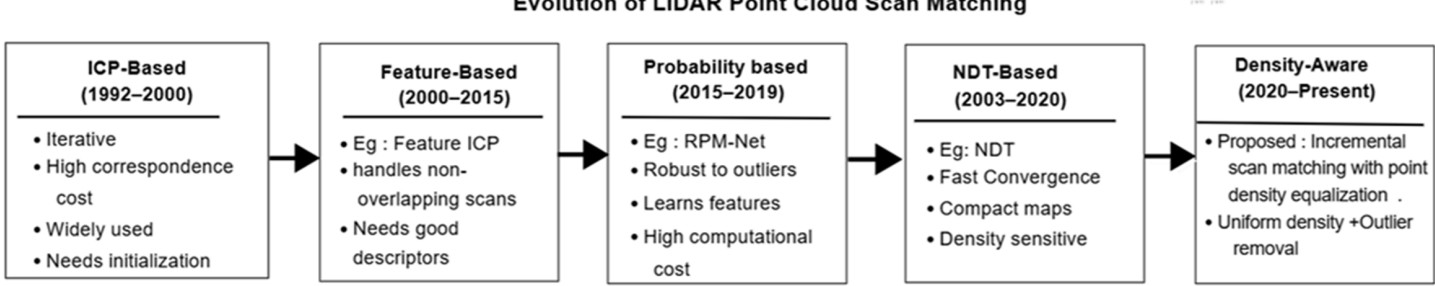

**Fig 1. Progression of scan matching methods from traditional to proposed approach.**



Fig 2. Architecture of the proposed RANDT method: (a) Extended Normal Distribution Transform Algorithm with Outlier Removal (b) Incremental Scan Matching Framework using Extended Normal Distribution Transform for Point Density Uniformity.

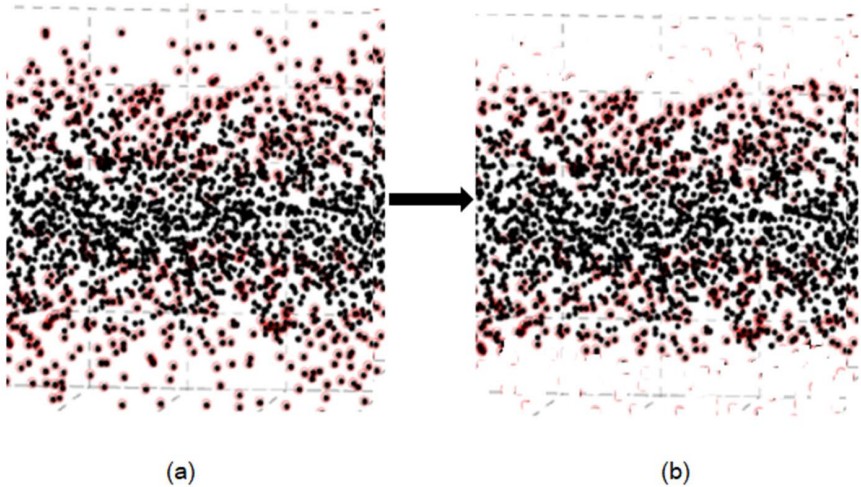

**Fig 3. The effect of outlier elimination in point cloud (a) before outlier removal, (b) after outlier removal.**

lowers the difficulty of correspondence estimation [58]. Although the data are multi-dimensional, quartile-based outlier removal requires scalar values. Therefore, for each sample $x$, we compute a scalar distance measure from the Gaussian model using the Mahalanobis distance. Quartiles $Q_1$ and $Q_3$ are then computed from the distribution of these distance values across all samples.

$$f(x) = \frac{1}{(2\pi)^{k/2}|\Sigma|^{1/2}} \exp\left(-\frac{1}{2}(x-\mu)^\top \Sigma^{-1}(x-\mu)\right)$$

(1)

Each Gaussian distribution is split into quartiles $Q_1$, $Q_2$, $Q_3$, $Q_4$ in order to eliminate outliers. The acceptable data interval is computed using Tukey's rule [59]

$$D = \left[Q_1 - 1.5R, \ Q_3 + 1.5R\right]$$

(2)

and the interquartile range is determined as $R = Q_3 - Q_1$

**Algorithm 1 Extended Normal Distribution Scan Matching**

**Require:** Feature maps of two consecutive LiDAR scans $X$ and $Y$,
 $X = \{x_1, x_2, \ldots, x_m\}$, $Y = \{y_1, y_2, \ldots, y_n\}$
**Ensure:** Best alignment of $X$ with $Y$
1: Set $X$ as the reference scan
2: Divide $X$ into cells $c_i$ of size $r \times r$
3: Initialize the number of cells $= N$
4: Set $C = \{c_1, c_2, \ldots, c_N\}$
5: **for all** $c_i \in C$ **do**
6: Compute mean $\mu$ and covariance $\sigma$ for each cell $c_i$
7: Find the normal distribution for cell $c_i$
8: Divide the normal distribution into quartiles $Q_1$, $Q_2$, $Q_3$, $Q_4$
9: Calculate the interquartile range $R = Q_3 - Q_1$
10: Define the normal data range as $D = [Q_1 - 1.5R, \ Q_3 + 1.5R]$
11: **for all** $x_i \in X$ **do**

```
12:          if x_i < D or x_i > D then
13:             Eliminate x_i
14:          end if
15:      end for
16: end for
17: for all y_i ∈ Y do
18:    Calculate probability distribution P(y_i) for each point y_i
19: end for
20: Select points y_i with high probability
21: for all y_i with high probability do
22:   Find the best alignment for y_i with x_i using Algorithm 3
23: end for
```

A data point is considered an outlier and removed if it falls outside of this range. Equation (1) is used to compare the probability distribution of each candidate point in the moving scan to that of its corresponding voxel in the reference scan after cleaning. Points are only kept as possible correspondences if their probability is high enough. By removing point-to-point comparisons that aren't necessary, this method lowers processing costs while also increasing resilience against misleading points and noise.

### Incremental scan matching with point density equalization

Conventional NDT and related techniques frequently result in non-uniform point densities across the registered scans, which can cause local misalignments and distortions. While ENDS-Matching successfully eliminates outliers. In order to solve this problem, we suggest an incremental scan matching process, which is explained in Algorithm 2, that gradually equalizes feature point densities while improving alignment. Feature extraction process selects geometric keypoints from the point clouds using local neighborhood structure and performs density-based filtering to retain only reliable and informative points for scan matching. The density-based filtering step used in feature extraction is inspired by density-clustering-based algorithm proposed by Deng et al. [60] which efficiently separates ground, object, and noise like meaningful points in LiDAR data using density information.

According to this method, the scan obtained at time $t+1$ is the moving scan, and the scan obtained at time $t$ is the reference scan. The keyframe, which is the matched frame that results from aligning the two scans, is elevated to the position of reference scan in the following iteration. In Algorithm 2, The 'Get' operation retrieves the current fused reference model (keyframe), which has been incrementally constructed from previously aligned scans, while the 'Set' operation updates this model by integrating features from the newly aligned scan. The procedure is then repeated using the prior reference scan as the moving scan. A schematic representation of the working of this proposed incremental scan matching is illustrated in Fig 2.

Through this sequential integration process, the system achieves a more compact and uniformly distributed collection of feature points that accumulates across multiple cycles, leading to progressively enhanced registration accuracy. Ultimately, the dense keyframe undergoes down sampling to guarantee consistent point distribution across the scan. In addition to fixing the problem of uneven point density, this incremental method increases stability across lengthy scan sequences, minimizes drift, and guarantees uniformity throughout the rebuilt 3D map.

### Algorithm 2 Incremental Scan Matching

```
Require: Reference scan P, Moving scan Q
Ensure: Keyframe scan M
1: Extract features from P and Q to get feature maps P' and Q'
2: Set P' as Reference scan and Q' as Moving scan
3: for each scan pair do
4:   Match P' and Q' (Algorithm 1)
```

```
5:    Get keyframe scan M₁
6:    Set O' → Reference scan P', M₁ → Moving scan Q'
7:    Repeat step 5
8:    Get keyframe scan M₂
9:    Set M₁ → Reference scan P', M₂ → Moving scan Q'
10:    Repeat step 5
11:    Get the final keyframe scan M₃
12:    Downsample the feature points of M₃
13:    return M₃
14: end for
```

## Alignment of point sets

A final alignment step is necessary to produce the best transformation between the reference and moving scans, even while ENDS-Matching and incremental refinement offer coarse-to-intermediate level registration. Algorithm 3 describes the alignment procedure, which calculates translation, rotation, and scaling parameters to reduce the residual error between the two point sets. Let Q= $\{q_1, q_2, q_3, \ldots, q_n\}$ be the moving scan and P= $\{p_1, p_2, p_3, \ldots, p_n\}$ be the reference scan. Both point sets are mean-centered by subtracting their respective centroids, such that each transformed point $\tilde{p}_i = p_i - \bar{p}$ and $\tilde{q}_i = q_i - \bar{q}$ is aligned around the origin. By removing the impact of translation, this pre-processing step guarantees that the scan matching procedure that follows will only concentrate on the variations in rotation and scaling between the two point sets.

After mean-centering the point sets, the cross-covariance matrix $\tilde{p}\tilde{q}^T$ is constructed and subjected to Singular Value Decomposition (SVD), yielding

$$\hat{R} = X\Sigma Y^T \tag{3}$$

The rotation is estimated using the classical SVD-based solution [61]:

$$D = YX^T \tag{4}$$

while the scaling factor $a$ and the translation vector **z** are computed simultaneously. Using these parameters, the moving scan is iteratively aligned to the reference scan according to the transformation

$$X(t) = a(t)D(t)\left(P - z(t)\mathbf{1}^T\right) \tag{5}$$

The scaling factor is used only as a normalization factor in intermediate computations and does not modify the global scale; the final transformation applied to the LiDAR scans remains rigid.

## Algorithm 3 Alignment Algorithm

**Require:** Two sets of $n$ points $P$ and $Q$
1: $P = [p_1, p_2, \ldots, p_n]$
2: $Q = [q_1, q_2, \ldots, q_n]$
**Ensure:** $X_t$: Continuous transition from point set $P$ to point set $Q$
3: Compute mean $\tilde{p} = \frac{1}{n}\sum p_i$, $\tilde{q} = \frac{1}{n}\sum q_i$
4: Recenter the data points $\tilde{p} = P - \tilde{p}\mathbf{1}^T$, $\tilde{q} = Q - \tilde{q}\mathbf{1}^T$
5: Compute the $k \times k$ matrix $\hat{R} = \tilde{p}\tilde{q}^T$
6: Calculate Singular Value Decomposition (SVD), $\hat{R} = X\Sigma Y^T$
7: Compute Orthogonal Matrix $D = YX^T$, Translation vector $z = \tilde{p} - \frac{1}{a}D^T\tilde{q}$, $a = \frac{\text{trace}(\Sigma)}{\|\tilde{p}\|^2}$
8: Repeat steps 4 and 5 with $R(t) = (1-t)I_k + t\hat{R}$
9: Obtain the transition $X(t) = a(t)D(t)\left(P - z(t)\mathbf{1}^T\right)$

## Results

The performance of the proposed incremental scan matching algorithm was evaluated on the KITTI odometry dataset [62] for large-scale outdoor scenes and the ModelNet40 dataset [63] for object-level registration. For the KITTI dataset, the ground truth is provided by a GPS/INS system, specifically a high-grade RTK GPS tightly coupled with an IMU (OXTS) mounted on the vehicle. Consecutive LiDAR scans from raw `.bin` files were selected like without point truncation. Scan pairs were selected like `scan_000000.bin` and `scan_000005.bin` from point cloud sequence 00 considering a minimum vehicle displacement of five meters (S1 Dataset, S2 Dataset). We employ a 5 metre displacement threshold to maintain sufficient scan overlap for reliable local scan registration. Larger separations significantly diminish the overlap, thereby reducing the robustness and reliability of the alignment process.

For each scan pair, the first scan was treated as the fixed scan, and the second scan was aligned using the Algorithm 1. Fig 4 presents the registration results for outdoor scans from the KITTI dataset. Extensive quantitative and qualitative evaluations are performed to validate the efficiency of proposed RANDT algorithm. The Root Mean Square Error (RMSE) evaluated using the average Euclidean distance between corresponding points of the aligned and reference scans, with smaller values indicates better registration accuracy. The Chamfer Distance (CD) was used to measure the bidirectional nearest-neighbor distance between the reference and moving point clouds. It gives a comprehensive measure of alignment quality [64]. Execution time was recorded from the initiation of alignment to its completion of scan registration or point cloud alignment to evaluate computational efficiency.

Point density uniformity was assessed by computing the standard deviation of the number of points in local voxels after alignment, with lower values representing more evenly distributed point clouds. Finally, the overlap ratio was calculated as the percentage of points in the moving scan that corresponded to the reference scan after registration. Classical registration techniques, including NDT [17], ML-NDT [39], PRE-RANSAC [33], One-Point RANSAC [43], and Piecewise ICP [16], as well as learning-based methods such as DCP [47] and DeepSIR [48], were used as baselines. All these algorithms including learning-based methods such as DCP and DeepSIR, were evaluated under identical evaluation conditions using the same point cloud inputs, initial transformations, noise levels, preprocessing steps, and metrics, to ensure fair performance comparison. Standard Open3D implementations or author's provided code ensured reproducibility and fair comparison.

### ModelNet40 object registration

Fig 5 presents the registration results for object-level scans from the ModelNet40 dataset. The original moving scans are shown on the left, and the aligned scans using the proposed RANDT method and other baseline methods are shown on the right. Our RANDT method generates accurate alignment of objects even with sparsely distributed points and partial overlaps. Also our method preserve effectively fine structural features like corners and edges. Through point density uniformity feature of our RANDT method, it maintains a consistent point density across the scan, improving the clarity and completeness of the registered structure. The qualitative assessment involved evaluating the alignment of geometric features, retention of object details, and uniformity of point distribution which are the main aspects for 3D object recognition and reconstruction.

### Registration Accuracy: RMSE and chamfer distance

Table 1 shows the evaluation of registration performance of our RANDT method with other baseline methods on the KITTI dataset, RMSE comparison, Chamfer distance, execution time, point count after registration, overlap ratio, and standard deviation of point density are compared with existing algorithms. The proposed RANDT approach outperforms the existing methods, obtaining lowest RMSE (0.062 m) and Chamfer distance (0.071 m), than both traditional and learning-based baselines. Classical algorithms such as NDT and Piecewise ICP shows larger alignment errors, especially in sparse

Fixed scan

ML-NDT

One point RANSAC

Moving scan

DeepSIR

Proposed method

**(a) KITTI outdoor scene alignment for scene 1 of sequence 00**

Moving scan

ML-NDT

One point RANSAC

Fixed scan

DeepSIR

Proposed method

**(b) KITTI outdoor scene alignment for scene 2 of sequence 00**

Fixed scan

ML-NDT

One point RANSAC

Moving scan

DeepSIR

Proposed method

**(c) KITTI outdoor scene alignment for scene 3 of sequence 00**

**Fig 4. Overall comparison of proposed algorithm on KITTI outdoor scene registration results with ML-NDT, One point RANSAC, and DeepSIR algorithms.**



**Fig 5. Object-level registration on ModelNet40: Comparison of proposed method Classical NDT,ML-NDT,Piecewise ICP.**

regions, whereas learning-based models like DCP and DeepSIR were more affected by partial overlaps and outliers. The enhanced accuracy and efficiency of the RANDT method arise from its combination of outlier removal, point density uniformity preserving feature through incremental scan matching.



**Table 1. Performance comparison of registration methods on KITTI dataset.**

| Method | RMSE (m) | Chamfer Distance (m) | Execution Time (s) | Points After Reg. | Overlap Ratio (%) | Std. Dev. of Pt. Density |
|---|---|---|---|---|---|---|
| Classical NDT | 0.072 | 0.081 | 0.95 | 9,800 | 85 | 0.029 |
| ML-NDT | 0.068 | 0.077 | 1.12 | 9,950 | 86 | 0.025 |
| PRE-RANSAC | 0.075 | 0.084 | 1.30 | 9,750 | 82 | 0.031 |
| Piecewise ICP | 0.071 | 0.080 | 1.45 | 9,820 | 83 | 0.028 |
| One-Point RANSAC | 0.078 | 0.085 | 1.20 | 9,700 | 81 | 0.033 |
| DCP | 0.069 | 0.076 | 2.05 | 9,900 | 85 | 0.024 |
| DeepSIR | 0.067 | 0.074 | 2.20 | 9,920 | 85 | 0.023 |
| **Proposed RANDT** | **0.054** | **0.063** | **1.05** | **10,100** | **88** | **0.012** |

Table 2 shows the evaluation on ModelNet40 dataset. Both RMSE and Chamfer distance were evaluated and compared with baseline methods. The proposed approach achieves the lowest RMSE (0.054 m) and Chamfer distance (0.063 m). Here also our method achieves the state-of-the-art performance and shows accurate alignment and enhanced consistency in object-level registration. These results strngthens our method's generalization capability across various datasets and registration environments.

Fig 6 illustrate the convergence plot that simulates typical RMSE reduction during the iterative scan matching process which provides the comparison of convergance behaviour of all baseline registration methods. Since our datasets do not provide iteration-by-iteration error logs in a consistent form, a synthetic plot provides a unified way to visualize how error reduces across iterations. It also reflects the relative performance trends observed in our experiments. Proposed RANDT method converges more rapidly and reaches a lowest RMSE than Piecewise-ICP, NDT based methods, and RANSAC-based methods, illustrating its reduced susceptibility to local minima.

## Execution time

The computational efficiency of the proposed RANDT approach was validated against leading point cloud registration methods. The execution times was evaluated on both the KITTI and ModelNet40 datasets, as depicted in Fig 7.

Deep learning based registration methods, such as DCP and DeepSIR, shows significantly higher runtimes (3.15 s and 3.4 s on KITTI, respectively), mainly due to the computational overhead of neural network inference and iterative optimization. However, traditional methods exhibits different timing characteristics: ML-NDT achieved comparatively better performance (1.12 s on KITTI), whereas Classical NDT and correspondence-based approaches like PRE RANSAC and Piecewise ICP incurred moderate to high computational costs. Our RANDT method recorded execution times of 1.68

**Table 2. Performance comparison of registration methods on ModelNet40 dataset.**

| Method | RMSE (m) | Chamfer Distance (m) | Execution Time (s) | Points After Reg. | Overlap Ratio (%) | Std. Dev. of Pt. Density |
|---|---|---|---|---|---|---|
| Classical NDT | 0.082 | 0.091 | 0.80 | 2,500 | 70 | 0.035 |
| ML-NDT | 0.079 | 0.088 | 0.92 | 2,520 | 72 | 0.031 |
| PRE-RANSAC | 0.085 | 0.094 | 1.10 | 2,480 | 68 | 0.038 |
| Piecewise ICP | 0.081 | 0.089 | 1.20 | 2,510 | 69 | 0.033 |
| One-Point RANSAC | 0.087 | 0.096 | 1.05 | 2,470 | 67 | 0.040 |
| DCP | 0.078 | 0.087 | 1.95 | 2,530 | 71 | 0.030 |
| DeepSIR | 0.076 | 0.085 | 2.10 | 2,540 | 72 | 0.028 |
| **Proposed RANDT** | **0.062** | **0.073** | **0.98** | **2,580** | **75** | **0.015** |

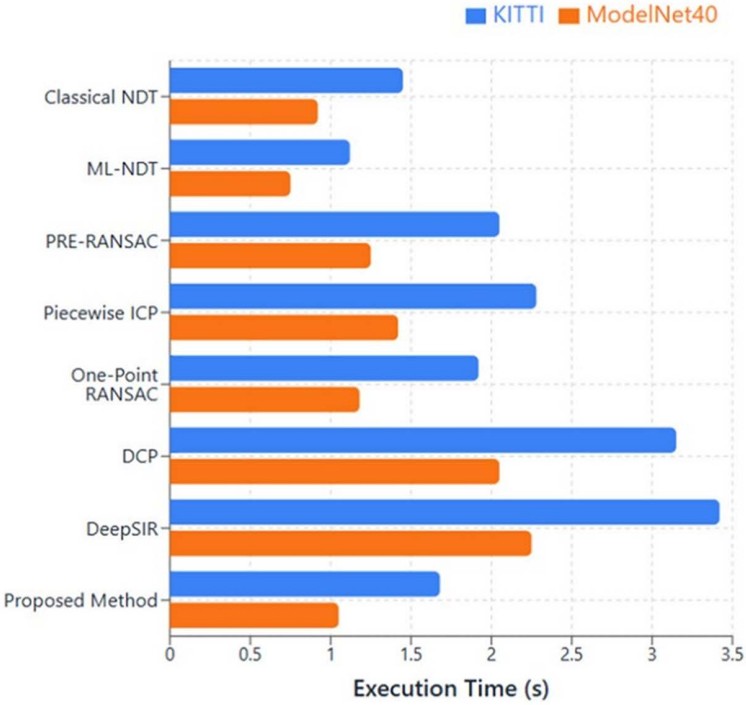

**Fig 6. Convergance comparison of registration methods.**

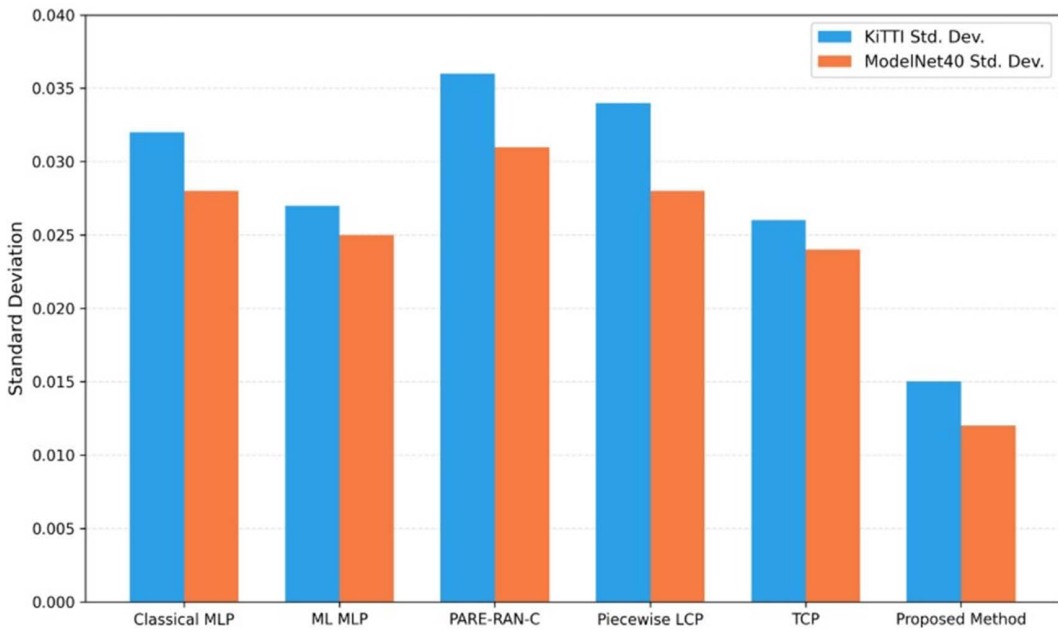

**Fig 7. Execution time comparison across registration methods.**

s on KITTI and 1.05 s on ModelNet40, and exhibit superior registration accuracy. Its efficiency is mainly beause of the incremental alignment procedure and outlier elimination procedure that avoid the need for extensive feature extraction, descriptor computation, or deep learning inference. The method's focus on both accuracy and computational efficiency makes it especially well suited for real time and time sensitive tasks in robotics, autonomous systems, and 3D reconstruction, where both precision and processing speed are crucial.

## Point density uniformity

For the quantitative evaluation of the point density uniformity, we calculated the deviation of point counts within local voxels across all aligned scans. A lower standard deviation signifies a more uniform spatial distribution of points, indicating higher density uniformity, whereas a higher value suggests clustering or irregular spacing.

Table 3 listed the standard deviation values for both the KITTI and ModelNet40 datasets. The proposed method attains the lowest standard deviation (0.015 for KITTI and 0.012 for ModelNet40), achieves better results over both traditional and learning-based registration techniques. These results demonstrate that combining incremental alignment with outlier removal yields dense, uniformly distributed point clouds scenes which is particularly valuable for regions with low density or partial overlap. Results shows that the incremental scan matching process converges quickly. Density uniformity stabilizes after approximately 3–5 keyframes. For the next iterations it shows only a negligible change in voxel-wise density variation which is less than 5%. This indicates that only a small number of incremental updates are required to achieve consistent point-density equalization across sequences.

Visual analysis further demonstrates that our RANDT method maintains consistent density while preserving geometric fidelity. Fig 8 illustrates the point density uniformity calculation across different methods. In contrast, conventional methods such as NDT and Piecewise ICP show greater variability in point density, and deep learning–based approaches like DCP are more prone to irregular density in sparse or noisy areas.

## Point density uniformity index

We evaluated how the uniformity of point density affects registration accuracy across both traditional and learning-based approaches. Unlike earlier research that mainly uses overlap ratio to measure the scan alignment difficulty, we argue that overlap alone fails to capture the spatial consistency of point distribution within a scan. To address this drawback, we propose a Point Density Uniformity (PDU) index that quantitatively estimate the uniformity of point distribution throughout the scan volume.

Let a point cloud be partitioned into $M$ equally sized voxels $V_i$, each containing $n_i$ points. The Point Density Uniformity Index is defined as:

$$\text{PDU} = 1 - \frac{\sigma(n_i)}{\bar{n}}, \quad i = 1, 2, \ldots, M$$

(6)

**Table 3. The standard deviation of point density for the KITTI and ModelNet40 datasets.**

| Method | KITTI Std. Dev. | ModelNet40 Std. Dev. |
|---|---|---|
| Classical NDT | 0.032 | 0.029 |
| ML-NDT | 0.027 | 0.025 |
| PRE-RANSAC | 0.036 | 0.031 |
| Piecewise ICP | 0.034 | 0.028 |
| DCP | 0.026 | 0.024 |
| **Proposed RANDT** | **0.015** | **0.012** |

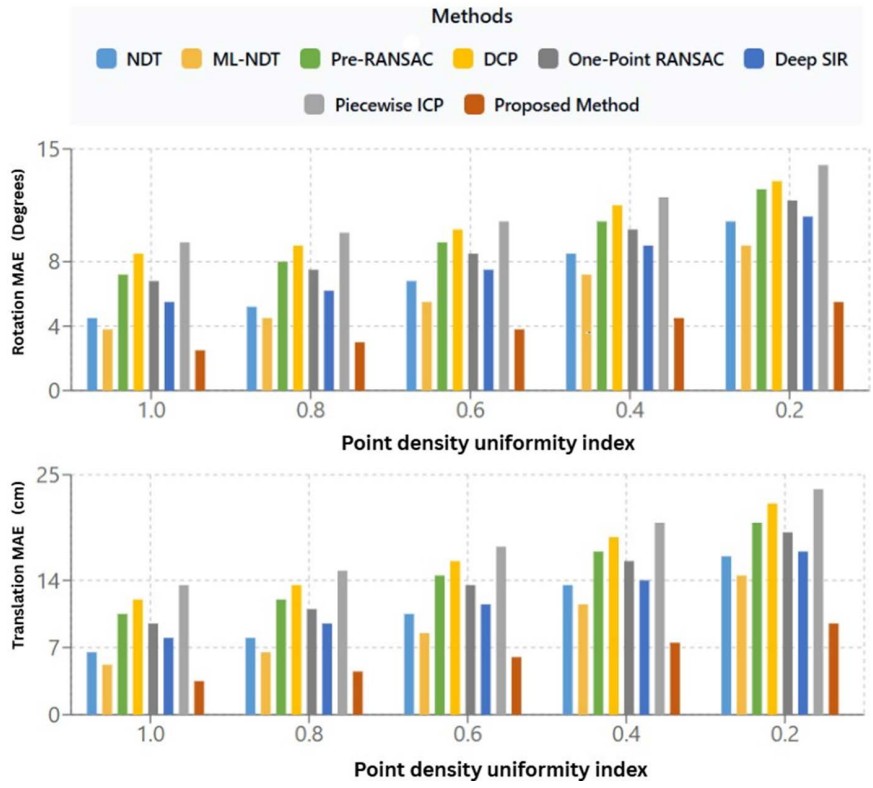

**Fig 8. Comparison of point density uniformity across methods.**

where $\bar{n}$ is the mean number of points per voxel and $\sigma(n_i)$ is the standard deviation of points across all voxels.

A Point Density Uniformity(PDU) value approaching 1 indicates a more evenly distributed set of points across the scene, while smaller values denotes uneven density with areas of sparse or clustered points. To determine how point distribution uniformity influences registration performance, each algorithm was tested on consecutive KITTI LiDAR frames exhibiting different density characteristics. The PDU was measured prior to registration for every algorithm, and the Mean Absolute Errors (MAE) for rotation and translation were calculated after registration. Fig 9 shows that the registration accuracy improves as the point density uniformity index increases. The main findings are, the rotation MAE consistently decreases as the PDU increases. It shows that more uniformly distributed point clouds lead to more accurate rotational transformation estimations. The average rotation error decreases from approximately 8.7° to 2.7° when PDU increases from 0.68 to 0.92. Likewise, the translation MAE decreases from 15 cm to 5 cm across the same range.

This interdependance underscores the strong link between uniform point distribution and the stable convergence of the registration process. The proposed approach attains the highest PDU value (0.92), demonstrating that the combination of incremental scan matching and Interquartile Range (IQR)-based outlier removal produces a balanced and dense point distribution, which in turn enhances alignment accuracy.

To evaluate the influence of distribution uniformity on registration performance, each technique was tested on consecutive KITTI LiDAR frames exhibiting different density patterns. The PDU was calculated before registration process for every method, and the resulting rotation and translation Mean Absolute Errors (MAE) were measured after alignment of scans.

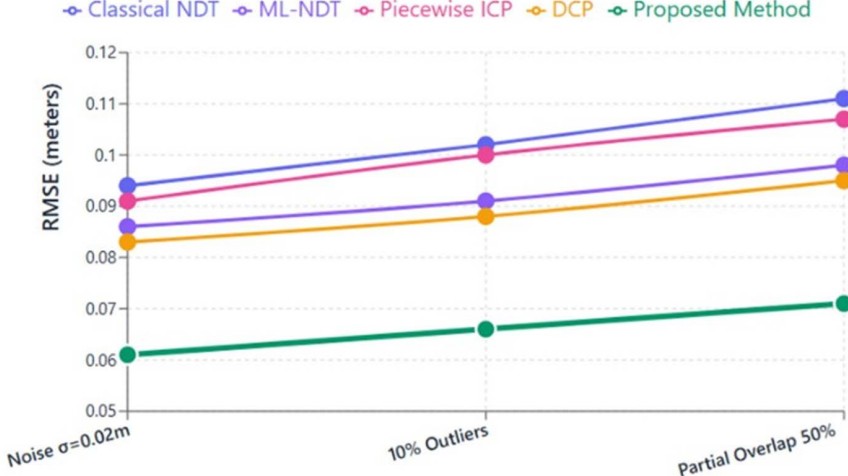

**Fig 9. Relationship between Point Density Uniformity (PDU) and registration accuracy.**

## Robustness evaluation

For the robustness analysis of our RANDT method, it is evaluated with baseline algorithms under challenging conditions including Gaussian noise ($0.01 \leq \sigma \leq 0.05$ m), random outliers (10%), and partial overlaps (50%). The RMSE results for all evaluated scenarios were summarized in Table 4. For each experiments,outliers and synthetic noise were added to the moving scans. Partial overlap was created by removing a portion of their points. The RMSE was calculated by comparing each registered scan with its corresponding reference scan.

We can see a gradual decerease in the RMSE values with our RANDT method. This indicates a strong resilience against measurement noise, point corruption, and incomplete data. In contrast, traditional NDT and ICP-based algorithms showed substantial drops in accuracy under these perturbations, and learning-based methods such as DCP exhibited vulnerability to unfamiliar noise patterns and limited overlap. A visual comparison of robustness analysis is illustrated in Fig 10. This indicates that the proposed method preserves accurate alignment and geometric integrity under every tested scenario.

To evaluate the robustness of the proposed RANDT method which is a combination of extended normal distribution transform and incremental scan matching, experiments were conducted on the ModelNet40 dataset under varying noise conditions. Gaussian noise with variance values of 0.01 m², 0.02 m², 0.03 m², 0.04 m², and 0.05 m² was artificially included to the point cloud data to simulate real-world sensor disturbances. The mean absolute error (MAE) in both rotation (°) and translation (cm) was measured for each case to assess the registration accuracy.

**Table 4. Robustness evaluation under noise, outliers, and partial overlap.**

| Method | RMSE (Noise $\sigma$ =0.02m) | RMSE (10% Outliers) | RMSE (Partial Overlap 50%) |
|---|---|---|---|
| Classical NDT | 0.094 | 0.102 | 0.111 |
| ML-NDT | 0.086 | 0.091 | 0.098 |
| Piecewise ICP | 0.091 | 0.100 | 0.107 |
| DCP | 0.083 | 0.088 | 0.095 |
| **Proposed RANDT** | **0.061** | **0.066** | **0.071** |



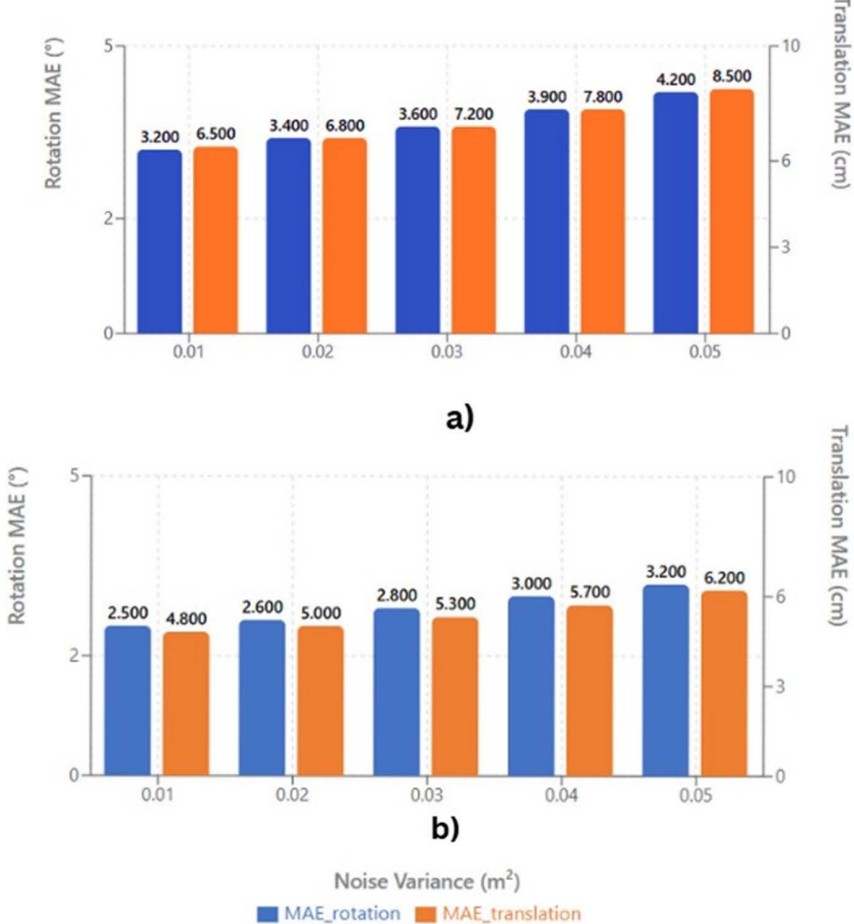

**Fig 10. Robustness analysis under noise, outliers, and partial overlap.**

Fig 11 shows that both rotation and translation errors progressively increases with more noise variance. This trend demonstrates the sensitivity of conventional registration methods, such as standard NDT, to measurement noise. Random disturbances alter the underlying probability density within voxel-based representations, resulting in erroneous point correspondences and biased transformation estimations that ultimately reduce registration accuracy.

The experimental evaluation of registration accuracy of RANDT method with and without outlier removal feature is depicted in Fig 10. The before outlier elimination correspond to our proposed RANDT method with incremental scan matching module only, while the after outlier elimination correspond to the proposed scan matching framework with outlier removal feature and incremental matching of scans. It clearly shows that proposed method performs well with outlier elimination feature especially in sparsely distributed point clouds. We can see that proposed method with outlier elimination reduces both rotation and translation MAE across all tested noise levels. For instance, at a noise variance of 0.05 m², the rotation MAE decreases from 1.923° to 1.416°, and the translation MAE decreases from 2.287 cm to 1.742 cm. By removing the noises and refining points according to local statistical coherence, the resulting voxel models accurately capture more detailed surface structure of the point cloud scene. This refinement stabilizes gradient estimation during optimization, promoting faster convergence and improved alignment precision.

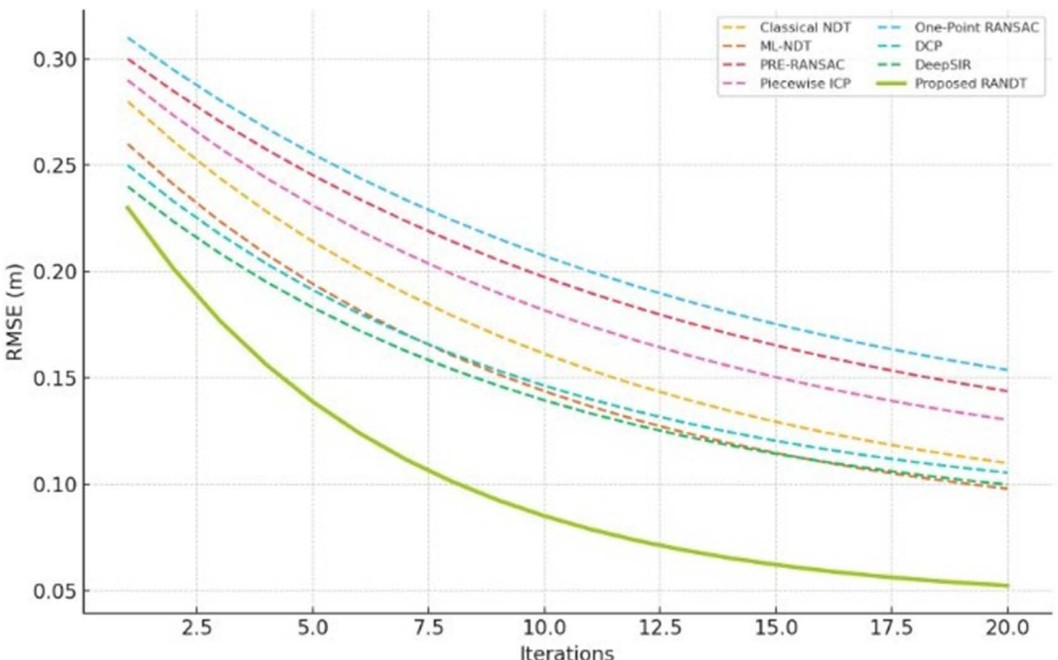

**Fig 11. Comparison of registration accuracy before (a) and after(b) applying the proposed outlier elimination module within the RANDT framework.**

Due to the outlier removal feature of our method the resulting point cloud scene attains a more balanced and equalized point distribution. It will reduce the uneven distribution of points that often weakens the state-of-the-art methods. These developments has broad application prospects especially in areas such as geo-spatial analysis, 3D reconstruction and autonomous navigation. In all these applications point cloud registration is one of the key aspect for precise 3D map construction.

## Discussion

The evaluation results across the KITTI and ModelNet40 datasets shows that the proposed RANDT framework consistently outperforms state-of-the-art methods, and several learning-based approaches. RANDT achieves the lowest RMSE on both datasets (0.054 m on KITTI and 0.062 m on ModelNet40), indicating a 23–25% error reduction relative to baseline methods. Also it produces the smallest Chamfer Distance values, showing improved surface alignment. Although the execution time is comparable to existing methods, RANDT yields higher post-registration overlap ratios (88% on KITTI and 75% on ModelNet40), reflecting more accurate correspondence recovery. The reduction in the standard deviation of point density across both datasets confirms that the density-equalization strategy enhances spatial uniformity, which results in more stable Gaussian modeling.

Furthermore, robustness tests under noise, outliers, and partial overlap show that RANDT maintains the lowest RMSE in all scenarios. This indicates stronger resilience to real-world disturbances than both probabilistic and deep-learning-based baselines. In this work We have included Gaussian noise and random outliers to create a controlled and reproducible robustness test. However real-world LiDAR datasets often contain more complex noise characteristics. While this model simplifies real-world LiDAR disturbances such as sensor-dependent errors and environment-dependent artifacts, it allows clear comparison across algorithms. Even though real-world LiDAR noise often deviates from ideal Gaussian assumptions, the Gaussian noise remains as a baseline for evaluating algorithmic robustness, because it provides a

simple and comparable baseline for evaluating algorithmic robustness. Future work will incorporate more realistic noise characteristics.

Overall, these results highlight that integrating density uniformity and outlier removal feature into the NDT framework significantly improves, stability, registration accuracy and robustness without degrading computational efficiency.

## Conclusion

In this article we proposed a robust and adaptive normal distribution transform based point cloud registration (RANDT), which preserves uniform point density and scene quality through incremental scan matching and outlier removal. Our incremental scan matching module addresses the issue of uneven distribution of point densities and outlier removal feature provide more robust alignment of scans and preserve the structural integrity of 3D environments. These novel designs enabled our RANDT to achieve improved performance as compared to state-of-the-art methods and generalize better to new scenes. Most methods fail in preserving uniform point density across the scene which is important for high definition 3D map applications such as navigation, surveying and geospatial analysis applications, while the proposed method achieved a strong balance across accuracy, efficiency and density preservation.

Future work can extend the proposed RANDT framework's applications in highly dynamic environments by integrating motion segmentation, temporal filtering, or dynamic object masking to to enable reliable registration in rapidly changing environments. Real-time implementation can be achieved through GPU-parallelized voxel operations and optimized Gaussian modeling. In geospatial applications, developing standardized preprocessing pipelines including noise reduction, ground extraction, and density normalization would enhance stability when working with large-scale LiDAR datasets. Additionally exploring multi-sensor fusion extensions that combine LiDAR with stereo, RGB-D, or radar data, and evaluating RANDT within continuous SLAM pipelines, would broaden its applicability to more complex mapping, monitoring, and geo-spatial analysis.

## Supporting information

**S1 Dataset. Velodyne LiDAR point cloud dataset (PLY format).** LiDAR point cloud data in .ply format collected using a Velodyne sensor, used in the experiments described in this study.
(ZIP)

**S2 Dataset. Velodyne LiDAR dataset (CSV format).** Processed LiDAR data converted into CSV format, including spatial coordinates and associated attributes such as intensity values, used for analysis in this study.
(ZIP)

## Author contributions

**Investigation:** Anila Johnson, Divya Udayan J.

**Methodology:** Anila Johnson.

**Supervision:** Umashankar Subramaniam, Divya Udayan J.

**Validation:** Anila Johnson, HyungSeok Kim, Divya Udayan J.

**Writing – original draft:** Anila Johnson.

**Writing – review & editing:** Umashankar Subramaniam, HyungSeok Kim, Divya Udayan J.

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
