## [Decision Letter · Decision Letter 0]

1 Dec 2025

PONE-D-25-58849Density-equalized RANDT scan matching with integrated outlier removal and point density uniformityPLOS ONE

Dear Dr. J,

Thank you for submitting your manuscript to PLOS ONE. After careful consideration, we feel that it has merit but does not fully meet PLOS ONE’s publication criteria as it currently stands. Therefore, we invite you to submit a revised version of the manuscript that addresses the points raised during the review process.

We look forward to receiving your revised manuscript.

Kind regards,

Shih-Lin Lin, Ph.D

Academic Editor

PLOS ONE

Journal Requirements:

3. In the online submission form, you indicated that all relevant data will be made available upon demand.

5. Please update your submission to use the PLOS LaTeX template. The template and more information on our requirements for LaTeX submissions can be found at http://journals.plos.org/plosone/s/latex.

Reviewers' comments:

Reviewer's Responses to Questions

**Comments to the Author**

1. Is the manuscript technically sound, and do the data support the conclusions?

Reviewer #1: Partly

Reviewer #2: Partly

2. Has the statistical analysis been performed appropriately and rigorously? 

Reviewer #1: Yes

Reviewer #2: N/A

3. Have the authors made all data underlying the findings in their manuscript fully available?

The PLOS Data policy requires authors to make all data underlying the findings described in their manuscript fully available without restriction, with rare exception (please refer to the Data Availability Statement in the manuscript PDF file). The data should be provided as part of the manuscript or its supporting information, or deposited to a public repository. For example, in addition to summary statistics, the data points behind means, medians and variance measures should be available. If there are restrictions on publicly sharing data—e.g. participant privacy or use of data from a third party—those must be specified.requires authors to make all data underlying the findings described in their manuscript fully available without restriction, with rare exception (please refer to the Data Availability Statement in the manuscript PDF file). The data should be provided as part of the manuscript or its supporting information, or deposited to a public repository. For example, in addition to summary statistics, the data points behind means, medians and variance measures should be available. If there are restrictions on publicly sharing data—e.g. participant privacy or use of data from a third party—those must be specified.requires authors to make all data underlying the findings described in their manuscript fully available without restriction, with rare exception (please refer to the Data Availability Statement in the manuscript PDF file). The data should be provided as part of the manuscript or its supporting information, or deposited to a public repository. For example, in addition to summary statistics, the data points behind means, medians and variance measures should be available. If there are restrictions on publicly sharing data—e.g. participant privacy or use of data from a third party—those must be specified.requires authors to make all data underlying the findings described in their manuscript fully available without restriction, with rare exception (please refer to the Data Availability Statement in the manuscript PDF file). The data should be provided as part of the manuscript or its supporting information, or deposited to a public repository. For example, in addition to summary statistics, the data points behind means, medians and variance measures should be available. If there are restrictions on publicly sharing data—e.g. participant privacy or use of data from a third party—those must be specified.

Reviewer #1: No

Reviewer #2: Yes

4. Is the manuscript presented in an intelligible fashion and written in standard English?

Reviewer #1: Yes

Reviewer #2: Yes

5. Review Comments to the Author

Reviewer #1: 1. The abstract lacks quantitative performance results—such as specific accuracy improvements, convergence speed, or error reductions—making it difficult to assess the actual impact of the proposed RANDT framework compared with existing methods.

2. The writing is repetitive and somewhat unfocused; key contributions (e.g., incremental scan matching, outlier removal) are mentioned but not clearly distinguished, and several sentences contain grammatical issues that reduce clarity and readability.

3. The introduction should clearly conclude with a distinct section highlighting the novel contributions of your work.

4. The literature review should benefit from more explorations of previous studies.

5. The discussion section needs to be expanded to more thoroughly analyze the results.

6. The first paragraph of the conclusion should succinctly summarize the contributions of the study in past tense not in present from.

7. The second paragraph of the conclusion should provide clear and actionable future recommendations.

8. Some equations require proper references.

9. Figures should be added within the manuscript, its hard to review without them isolated.

Reviewer #2: 1.The motivation for addressing point density uniformity is clearly stated, but the introduction could further elaborate on how density imbalance specifically degrades downstream tasks. Could the authors provide concrete examples or citations quantifying these effects?

2.The Extended NDT with outlier removal is a central contribution, yet the statistical rationale for using the IQR method could be expanded. How does IQR thresholding compare with more adaptive or multi-scale outlier filters in LiDAR processing?

3.Algorithm 1’s description indicates voxel-wise quartile computation but does not clarify how quartiles are computed for multivariate Gaussian distributions. Clarifying whether quartiles refer to probability density values or individual coordinate distributions would improve reproducibility.

4.In Algorithm 2, the term “feature extraction” is mentioned, but no details are provided about which features are used. Are these geometric features, keypoints, or simply raw points filtered by density?

5.The incremental scan matching framework appears essential for density equalization. Could the authors quantify how many iterations or keyframes are typically required before uniformity stabilizes across sequences?

6.In the experiments, scans are selected with a 5 m displacement from KITTI, yet the paper does not justify why this threshold is appropriate. Would larger displacements (e.g., 10–20 m) affect the robustness of the proposed method?

7.The paper claims that RANDT avoids local minima more effectively than ICP-type methods. Could the authors provide convergence plots or visualizations demonstrating this behavior across multiple trials?

8.It is suggested to add articles entitled “Puriyanto et al. Ball Detection System for a Soccer on Wheeled Robot Using the MobileNetV2 SSD Method”, “Fahad et al. Efficient Object Detection with an Optimized YOLOv8x Model” and “Trujillo et al. Enhancing Trajectory Tracking in Humanoid Robots Using Neural Network-Based Dynamic Gain Control” to the literature review.

9.In the robustness evaluation, the authors introduce Gaussian noise and outliers. However, the noise model seems simplistic. Do real-world datasets exhibit noise characteristics that differ from the simulated Gaussian assumptions?

10.The scaling factor in Algorithm 3 suggests that the transformation allows isotropic scaling. Most LiDAR registrations assume rigid transformations (no scale). Could the authors justify including scaling and discuss its impact?

11.The comparison with learning-based methods (DeepSIR, DCP) is appreciated, but no mention is made of training conditions or dataset biases. Were the pretrained models guaranteed not to have seen the evaluation scenes?

6. PLOS authors have the option to publish the peer review history of their article (what does this mean?). If published, this will include your full peer review and any attached files.). If published, this will include your full peer review and any attached files.). If published, this will include your full peer review and any attached files.). If published, this will include your full peer review and any attached files.

...

Reviewer #1: No

Reviewer #2: No

---

## [Author Response · Author response to Decision Letter 1]

24 Dec 2025

Reply to Review Comments on Submission ID : PONE-D-25-58849 entitled, “Density-equalized RANDT scan matching with integrated outlier removal and point density uniformity”

Authors’ General Reply: We express our deepest gratitude to the Journal Editor and the anonymous reviews for sparing their invaluable time and giving such constructive feedback which has further uplifted the quality of our manuscript in its revised form.

As we can clearly see, your opinion followed a meticulous review of our earlier submission and therefore we have paid much attention to all your constructive suggestions. Changes made in response to the review comments are highlighted in yellow throughout the manuscript. We also offer a detailed list of all the changes made in our reply to the individual comments.

Hope you will find our revised work worthy enough for your consideration and acceptance. Should you have any other minor concern(s), we are open to including them to the extent possible in future submissions.

We have added HyungSeok Kim (Konkuk University, South Korea) as a co-author based on his contributions to validation and writing – review & editing. His details and contributions have been updated in the manuscript.

A detailed point-by-point response is provided below. Reviewer comments are reproduced in bold, followed by our responses in regular text. All changes are highlighted in the revised manuscript.

Authors’ Reply: We have carefully checked the PLOS ONE formatting guidelines and verified that our manuscript now follows the required style. The file names, manuscript structure, headings, references, and figure/table placement have been updated to conform fully to the PLOS ONE template. The revised submission adheres to the formatting specifications provided in the links to the best of my knowledge.

Authors’ Reply: We confirm that all author-generated code used in this study will be made publicly available without restriction upon publication, in accordance with PLOS ONE’s guidelines. The code will be deposited in an openly accessible repository Github together with documentation to support reproducibility.

3. In the online submission form, you indicated that all relevant data will be made available upon demand.

Authors’ Reply: In compliance with the journal’s data availability requirements, all datasets used in this study have been deposited in a publicly accessible GitHub repository.Link is given below. https://github.com/anilabibin/Point_Cloud_registration.git

The Data Availability Statement has been updated to reflect this.

Authors’ Reply: We confirm that we are able to comply with PLOS ONE’s open data policy. All derived data generated during our experiments, along with the author-generated code, will be deposited in a public repository GitHub and made freely accessible upon publication. The datasets we have used is already uploaded in GitHub and the link is provided. https://github.com/anilabibin/Point_Cloud_registration.git The Data Availability Statement has been updated accordingly to reflect this.

5. Please update your submission to use the PLOS LaTeX template. The template and more information on our requirements for LaTeX submissions can be found at http://journals.plos.org/plosone/s/latex.

Authors’ Reply: We thank the editorial team for the comment. We confirm that our manuscript has been prepared using the official PLOS ONE LaTeX template. We have now updated the title page and ensured that all required formatting elements follow the PLOS ONE template guidelines to the best of our knowledge.

Kindly let us know if any further formatting is required.

Authors’ Reply: We thank the editor for the clarification. We have reviewed all the publications suggested by the reviewers and found them relevant to our study. Accordingly, we have incorporated these citations into the revised manuscript at appropriate locations.

Reviewer #1

Comment 1:

“The abstract lacks quantitative performance results—such as accuracy improvements, convergence speed, or error reductions—making it difficult to assess the actual impact of the proposed method.”

Authors’ Reply: We have now included quantitative results summarising the improvement achieved by the proposed method. Specifically, the abstract now reports the error reduction of 18-25%, PDU(Point Density Uniformity score) improvement of proposed RANDT method (Page 1, Last sentence of abstract).

Deleted : Experimental results and comparison with state-of-the-art methods demonstrate that RANDT can achieve better performance. Moreover,the results demonstrate that traditional registration methods doesn't preserve uniform point distribution,while RANDT can still achieve satisfactory performance even with noisy datasets.

Added : The evaluations on KITTY and ModelNet40 datasets demonstrate that proposed RANDT method achieves RMSE values of 0.054 m and 0.062 m with an error reduction of 18-25\% even with noise and partial overlaps, and also attains highest point desity uniformity score 0.92 than other baseline methods

Comment 2:

The writing is repetitive and somewhat unfocused; key contributions (e.g., incremental scan matching, outlier removal) are mentioned but not clearly distinguished, and several sentences contain grammatical issues that reduce clarity and readability.

Authors’ Reply: We thank the reviewer for this valuable observation. We have substantially revised the manuscript to improve clarity and organization. Specifically, we (i) removed repetitive statements, (ii) clearly explained major contributions (incremental scan matching, outlier removal, and point-density uniformity enhancement), and (iii) corrected all grammatical inconsistencies throughout the text. These improvements enhance the readability and focus of the paper. The revisions are incorporated in the Abstract and Introduction sections (Pages 1-2).

Deleted : As far as we are aware, no existing studies on point cloud registration has focused on the problem of point density uniformity along with outlier removal which is essential for high definition three dimensional point cloud maps.

While many algorithms are developed for point cloud registration , Deleted : no methods focuses on maintaining uniform point density during scan registration and sensitivity to outliers indicates Added : no existing studies on point cloud registration has focused on the problem of point density uniformity along with outlier removal which is essential for high definition three dimensional point cloud maps

Deleted : Although many existing techniques focus on acceleration, noise reduction, or managing partial overlaps, relatively few address the issue of maintaining uniform point cloud density throughout the alignment process

In our method an incremental scan matching module is introduced Added : which continuously perform scan matching with newly matched scans to achieve uniform and dense point distribution across the scene. Prior to the scan matching process an outlier removal feature Added : which removes the noisy data points is also included to achieve more accurate point cloud data.

Comment 3:

The introduction should clearly conclude with a distinct section highlighting the novel contributions of your work.

Authors’ Reply: We have revised the end of the Introduction to include a clearly separated “Contributions” subsection that explicitly highlights the novel aspects of our work.(Page 3 just before the last paragraph of introduction )

Added : The integration of Incremental scan matching and IQR(Inter Quartile Range) based outlier removal provides a novel unified framework for the point cloud registration process which improves both registration accuracy and robustness. To the best of our knowledge, no existing method integrates all of these components within a unified point-cloud registration pipeline.

Also the main contributions are highlighted as follows:

• A variant of Normal distribution based scan matching algorithm which is designed to serve as a reliable platform for high definition LiDAR map applications.

• We developed an Incremental scan matching framework for preserving uniform point density across the scans, which is essential for producing high-quality three-dimensional LiDAR maps.

• An outlier removal method which improves the accuracy of point cloud matching and precision of point correspondences during the scan matching process

• An efficient algorithm that estimates the transformation parameters is provided for transformation of moving scan to the reference scan.

• The suggested RANDT method improves the accuracy of point cloud matching and precision of point correspondences compared to existing techniques.

• To the best of our knowledge, no prior work unifies these components within a single point-cloud registration pipeline.

Comment 4:

The literature review should benefit from more explorations of previous studies.

Authors’ Reply: We have substantially expanded the literature review across all subsections—Classical Methods, Feature-Based Methods, Probabilistic/Distribution-Based Methods, Robust Estimation, and Deep Learning–Based Registration—to provide broader coverage of recent developments in point cloud registration. We incorporated several new studies published between 2020–2025 that address multi-resolution registration, probabilistic modeling, robust estimation, and transformer-based deep networks.

Specifically, we added discussions on:

• Zou H, et al. (2025)

Application of 3D point cloud and visual–inertial data fusion in robot dog autonomous navigation. PLOS One. 2025;20(2):e0317371.

• Yang Y, Holst C. (2025)

Piecewise-ICP: Efficient and robust registration for 4D point clouds in permanent laser scanning. ISPRS Journal of Photogrammetry and Remote Sensing. 2025;227:481–500.

• Chen Y, et al. (2025)

Overlapping point cloud registration algorithm based on KNN and the channel attention mechanism. PLOS One. 2025;20(6):e0325261.

• Bash EA, Wecker L, Rahman MM, Dow CF, McDermid G, Samavati FF, et al. (2023)

A multi-resolution approach to point cloud registration without control points. Remote Sensing. 2023;15(4):1161.

• Ahmadli I, Bedada WB, Palli G. (2022)

Deep learning and OcTree-GPU-based ICP for efficient 6D model registration of large objects. In: Human-Friendly Robotics 2021. Springer; 2022. p. 29–43.

• Yang X, Wang H, Dong Z, Liu Y, Li Y, Yang B. (2024)

A novel method for registration of MLS and stereo reconstructed point clouds. IEEE Transactions on Geoscience and Remote Sensing. 2024;62:1–13.

• Luo Z, Shen T, Zhou L, Zhu S, Zhang R, Yao Y, et al. (2018)

GeoDesc: Learning local descriptors by integrating geometry constraints. In: ECCV. 2018. p. 168–183.

• Wang C, Yang Y, Shu Q, Yu C, Cui Z. (2020)

Point cloud registration algorithm based on Cauchy mixture model. IEEE Photonics Journal. 2020;13(1):1–13.

• Goel K, Michael N, Tabib W. (2023)

Probabilistic point cloud modeling via self-organizing Gaussian mixture models. IEEE Robotics and Automation Letters. 2023;8(5):2526–2533.

• Li H, Liu Y, Men C, Fang Y. (2021)

A novel 3D point cloud segmentation algorithm based on multi-resolution supervoxel and MGS. International Journal of Remote Sensing. 2021;42(22):8492–8525.

• Yang H, Shi J, Carlone L. (2020)

TEASER: Fast and certifiable point cloud registration. IEEE Transactions on Robotics. 2020;37(2):314–333.

• Huang S, Gojcic Z, Schindler K. (2021)

PREDATOR: Registration of 3D point clouds with low overlap. In: CVPR. 2021.

• Qin Z, Yu H, Wang C, Guo Y, Peng Y, Ilic S, et al. (2023)

GeoTransformer: Fast and robust point cloud registration with geometric transformer. IEEE Transactions on Pattern Analysis and Machine Intelligence. 2023;45(8):9806–9821.

• Yu H, Li F, Saleh M, Busam B, Ilic S. (2021)

CoFiNet: Reliable coarse-to-fine correspondences for robust point cloud registration. In: NeurIPS. 2021;34:23872–23884.

These additions strengthen the contextual foundation for our proposed method and clearly position our contributions within the broader landscape of recent research.

Added: Enhancements on classical point cloud registration methods focuses on accelerated ICP variants like point to plane multi-resolution ICP and GPU-ICP [25]. It is best suitable for large scale LiDAR datasets since it improves convergence speed [26]. To improve the robustness under partially overlapped or sparsely distributed scans, a hybrid geometric frameworks that integrate voxel hashing and adaptive correspondence filtering have also introduced. (Subsection “Classical methods”)

Some of the feature based registration approaches incorporate graph-structured descriptors, like GCN-based local embeddings and GeoDesc [32]. It can capture higher order geometric relationships which improves correspondence reliability. To achieve robust matching under viewpoint changes and significant point sparsity methods like 3D3 and YOHO employ orientation-aware feature learning and hierarchical neighbourhood aggregation [33]. (Subsection “Feature based methods”)

The advancements in NDT based scan registration including multi-resolution voxel distribution models and Gaussian mixture-regression–based registration improves robustness against non-uniform sampling [37,38]. Also the point cloud registration methods employing kernel adaptive weighting have shown improved accuracy in highly dynamic or cluttered urban environments. (Subsection “Probabilis

---

## [Decision Letter · Decision Letter 1]

1 Feb 2026

PONE-D-25-58849R1Density-equalized RANDT scan matching with integrated outlier removal and point density uniformityPLOS One

Dear Dr. J,

Thank you for submitting your manuscript to PLOS ONE. After careful consideration, we feel that it has merit but does not fully meet PLOS ONE’s publication criteria as it currently stands. Therefore, we invite you to submit a revised version of the manuscript that addresses the points raised during the review process.

We look forward to receiving your revised manuscript.

Kind regards,

Shih-Lin Lin, Ph.D

Academic Editor

PLOS One

**Journal Requirements:**

Reviewers' comments:

Reviewer's Responses to Questions

**Comments to the Author**

1. If the authors have adequately addressed your comments raised in a previous round of review and you feel that this manuscript is now acceptable for publication, you may indicate that here to bypass the “Comments to the Author” section, enter your conflict of interest statement in the “Confidential to Editor” section, and submit your "Accept" recommendation.

Reviewer #1: (No Response)

Reviewer #2: (No Response)

Reviewer #3: (No Response)

Reviewer #4: (No Response)

2. Is the manuscript technically sound, and do the data support the conclusions?

Reviewer #1: (No Response)

Reviewer #2: (No Response)

Reviewer #3: Yes

Reviewer #4: Yes

3. Has the statistical analysis been performed appropriately and rigorously? 

Reviewer #1: (No Response)

Reviewer #2: (No Response)

Reviewer #3: Yes

Reviewer #4: Yes

4. Have the authors made all data underlying the findings in their manuscript fully available?

The PLOS Data policy requires authors to make all data underlying the findings described in their manuscript fully available without restriction, with rare exception (please refer to the Data Availability Statement in the manuscript PDF file). The data should be provided as part of the manuscript or its supporting information, or deposited to a public repository. For example, in addition to summary statistics, the data points behind means, medians and variance measures should be available. If there are restrictions on publicly sharing data—e.g. participant privacy or use of data from a third party—those must be specified.requires authors to make all data underlying the findings described in their manuscript fully available without restriction, with rare exception (please refer to the Data Availability Statement in the manuscript PDF file). The data should be provided as part of the manuscript or its supporting information, or deposited to a public repository. For example, in addition to summary statistics, the data points behind means, medians and variance measures should be available. If there are restrictions on publicly sharing data—e.g. participant privacy or use of data from a third party—those must be specified.requires authors to make all data underlying the findings described in their manuscript fully available without restriction, with rare exception (please refer to the Data Availability Statement in the manuscript PDF file). The data should be provided as part of the manuscript or its supporting information, or deposited to a public repository. For example, in addition to summary statistics, the data points behind means, medians and variance measures should be available. If there are restrictions on publicly sharing data—e.g. participant privacy or use of data from a third party—those must be specified.requires authors to make all data underlying the findings described in their manuscript fully available without restriction, with rare exception (please refer to the Data Availability Statement in the manuscript PDF file). The data should be provided as part of the manuscript or its supporting information, or deposited to a public repository. For example, in addition to summary statistics, the data points behind means, medians and variance measures should be available. If there are restrictions on publicly sharing data—e.g. participant privacy or use of data from a third party—those must be specified.

Reviewer #1: (No Response)

Reviewer #2: (No Response)

Reviewer #3: Yes

Reviewer #4: Yes

5. Is the manuscript presented in an intelligible fashion and written in standard English?

Reviewer #1: (No Response)

Reviewer #2: (No Response)

Reviewer #3: Yes

Reviewer #4: No

6. Review Comments to the Author

Reviewer #1: Well responded, however, remove extra information from the abstract now please. it is a but extra in content.

Reviewer #2: The article has been revised very well, so I would suggest to accept in its present form.

The article has been revised very well, so I would suggest to accept in its present form.

Reviewer #3: Figures 4, 5, and 8 effectively illustrate the registration results. However, in Figure 8 (point density uniformity comparison), consider adding:

A color scale legend showing what the colors represent (point density values)

Quantitative labels on the visualizations showing actual density variance values

This would make the visual comparison more interpretable for readers.

Minor editorial suggestions

Page 1, Abstract: “desity uniformity” → “density uniformity” (typo in line 17 of abstract).

Page 9, Algorithm 2, Line 4: Consider rephrasing “for all Reference scan and Moving scan do” to “for each scan pair do” for clarity.

Page 22, Line 534: “solved the issue” → “addresses the issue” (to maintain consistency with present tense used elsewhere in conclusions).

References: Verify that all in-text citations match the reference list. I noticed some formatting inconsistencies (e.g., reference 28 vs 29).

Strengths of the manuscript

Clear methodological contribution: The combination of IQR-based outlier removal with incremental scan matching is well-motivated and technically sound.

Comprehensive evaluation: Testing on both outdoor (KITTI) and object-level (ModelNet40) datasets demonstrates generalizability.

Thorough comparison: Seven baseline methods provide strong context for the improvements.

Reproducibility: Code and data availability support reproducibility.

Responsive revisions: The authors have thoroughly addressed previous reviewer comments.

Recommendation: Minor Revisions

This is a solid technical paper that makes meaningful contributions to LiDAR point cloud registration. The proposed RANDT method is novel, the experimental validation is rigorous, and the results are convincing. The suggestions I’ve outlined above are intended to strengthen an already good manuscript rather than address fundamental flaws. I believe the paper will be suitable for publication in PLOS ONE after these minor revisions are addressed.

I commend the authors for their thorough response to previous reviews and for the careful experimental work presented here. This method has practical potential for autonomous navigation, 3D reconstruction, and geospatial analysis applications.

Reviewer #4: Summary: The paper's innovation lies in its novel synthesis of three existing techniques (quartile-based outlier rejection, point equalization using SLAM, and orientation refinement using SVD) to obtain a marked improvement in pose-determination accuracy as compared to other widely used lidar scan-matching methods.

Feedback:

* Grammar: Another proofreading pass is needed to clean the text.

> In particular, there are many issues related to punctuation and spacing. There should never be a space before a period or comma, and there should always be a space at the beginning of a sentence. The first example is in the abstract: "Transform (RANDT) .In"

> There are also a several other grammar and spelling issues. Some examples include "overcoome" and "It offer"

> One of the author names on the "Author Contributions" list on p. 22 is spelled differently than in the author list on p. 1

* Algorithm Clarity: Some aspects of the algorithm description remain unclear.

> One of the most important algorithmic clarity issues involves equations (1) and (2). A one-dimensional distribution has clearly defined quartiles, but the definition of Q is not clear for a multi-dimensional distribution; it is possible (again not clear) that the quartiles were derived from f(x) rather than the samples x, but this again does not make sense because f(x) is one sided and the description in (2) assumes a two sided distribution of points.

> With regard to the pseudocode for Algorithm 2, the feature extraction step (line 1) has now been described in the text with one added sentence, but the methodology is imprecise. Is there a paper you can cite that describes the "density-based filtering" method you've applied?

> A second regarding Algorithm 2 is the definition of the "get" and "set" operations (e.g. in lines 5 and 6). It seems to me these can't simply be replacement operations (e.g. arrow meaning "becomes equal to"), as I thought that the reference frame needed to be a fusion of multiple raw frames in order to implement the equalization scheme. Bottom line, I'm a strong believer in point equalization in lidar scan matching ... but I still have no idea how this aspect of your algorithm was implemented.

* Results: In describing the results, a missing detail is a statement of the source of the ground truth. For KITTI, was GPS/INS used as ground truth to compute RMSE?

7. PLOS authors have the option to publish the peer review history of their article (what does this mean?). If published, this will include your full peer review and any attached files.). If published, this will include your full peer review and any attached files.). If published, this will include your full peer review and any attached files.). If published, this will include your full peer review and any attached files.

...

Reviewer #1: No

Reviewer #2: No

Reviewer #3: **Yes:** Ruben Dario Cardenas EspinosaRuben Dario Cardenas EspinosaRuben Dario Cardenas EspinosaRuben Dario Cardenas Espinosa

Reviewer #4: No

---

## [Author Response · Author response to Decision Letter 2]

17 Feb 2026

Reply to Review Comments on Submission ID : PONE-D-25-58849 entitled, “Density-equalized RANDT scan matching with integrated outlier removal and point density uniformity”

Authors’ General Reply: We express our deepest gratitude to the Journal Editor and the anonymous reviewers for sparing their invaluable time and giving such constructive feedback which has further uplifted the quality of our manuscript in its revised form.

As we can clearly see, your opinion followed a meticulous review of our earlier submission and therefore we have given much attention to all your constructive suggestions. Changes made in response to the review comments are highlighted in yellow throughout the manuscript. We also offer a detailed list of all the changes made in our reply to the individual comments.

Hope you will find our revised work worthy enough for your consideration and acceptance. Should you have any other minor concern(s), we are open to including them to the extent possible in future submissions.

A detailed point-by-point response is provided below. Reviewer comments are reproduced in bold, followed by our responses in regular text. All changes are highlighted in the revised manuscript.

Reviewer #1:

Well responded, however, remove extra information from the abstract now please. it is a but extra in content.

Authors Response : Thank you. The abstract has been shortened by removing the extra explanatory sentences and simplifying the description of the registration requirement.

Deleted : For constructing high definition three-dimensional environmental models with focus on reducing noise disturbances and preserving uniform point density an efficient point cloud scan matching or registration is required.

Added : Accurate registration is required to reduce noise and maintain uniform point density.

Deleted : Outliers or noises also degrade the accuracy of the maps. This leads to variations in the density of point distribution throughout the matched point cloud scan.

(From Abstract)

Reviewer #2:

The article has been revised very well, so I would suggest to accept in its present form.

Authors Response : Thank you

Reviewer #3:

Figures 4, 5, and 8 effectively illustrate the registration results. However, in Figure 8 (point density uniformity comparison), consider adding:

A colour scale legend showing what the colours represent (point density values)

Quantitative labels on the visualizations showing actual density variance values

This would make the visual comparison more interpretable for readers.

Authors Response : Thank you for the helpful suggestion. We have revised Figure 8 to include a color legend clarifying the meaning of the bar colors, which improves the interpretability of the point density variance comparison.

Minor editorial suggestions

Comment 1

Page 1, Abstract: “desity uniformity” → “density uniformity” (typo in line 17 of abstract).

Authors Response : Thank you for pointing this out. The typo has been corrected.

Comment 2

Page 9, Algorithm 2, Line 4: Consider rephrasing “for all Reference scan and Moving scan do” to “for each scan pair do” for clarity.

Authors Response : Thank you. The wording has been updated as suggested for clarity.

Comment 3

Page 22, Line 534: “solved the issue” → “addresses the issue” (to maintain consistency with present tense used elsewhere in conclusions).

Authors Response : Thank you. The wording has been updated as suggested

Comment 4

References: Verify that all in-text citations match the reference list. I noticed some formatting inconsistencies (e.g., reference 28 vs 29).

Authors Response : Thank you. All in-text citations now match the reference list, and the references have been reformatted for consistency according to the journal style.

Reviewer #4

Comment 1

Grammar:

There are many issues related to punctuation and spacing. There should never be a space before a period or comma, and there should always be a space at the beginning of a sentence. The first example is in the abstract: "Transform (RANDT) .In"

Authors Response : Thank you for pointing this out. We have corrected the punctuation and spacing issues throughout the manuscript, including the example in the abstract.

Comment 2

There are also some other grammar and spelling issues. Some examples include "overcoome" and "It offer"

Authors Response : Thank you for highlighting the grammar and spelling issues. We have carefully proofread the manuscript and corrected these errors throughout the text.

Comment 3

One of the author names on the "Author Contributions" list on p. 22 is spelled differently than in the author list on p. 1

Authors Response : Thank you for noting this error. The author’s name was misspelled in the Author Contributions section on page 22 and has now been corrected to “Umashankar Subramaniam” to match the author list.

Comment 4

Algorithm Clarity: Some aspects of the algorithm description remain unclear.

One of the most important algorithmic clarity issues involves equations (1) and (2). A one-dimensional distribution has clearly defined quartiles, but the definition of Q is not clear for a multi-dimensional distribution; it is possible (again not clear) that the quartiles were derived from f(x) rather than the samples x, but this again does not make sense because f(x) is one sided and the description in (2) assumes a two sided distribution of points.

Authors Response : Thank you for pointing out this lack of clarity. We have revised the manuscript to clarify that quartiles for each sample and rewritten that the computed values are from a scalar Mahalanobis distance, and quartiles are derived from the distribution of these distance values across all samples. This explanation has been added in the text before Eq. (2).

Added: “Algorithm 1 introduces this extension, which is the Extented Normal Distribution Scan Matching(E-NDSM). The input reference scan is divided into uniform voxels in the 251 first step. The set of points in a voxel is compactly represented by its mean vector and 252 covariance matrix, creating a multivariate Gaussian distribution, rather than matching 253 every point within each voxel. For each voxel, quartiles are computed over the scalar 250 254 probability density values obtained from the multivariate Gaussian within each voxel, instead of computing voxels individually on x, y, and z coordinate dimensions. Hence it 256 provides a single scalar metric for outlier evaluation. Since the pair (µ,Σ) can statistically describe the entire voxel, this lowers the difficulty of correspondence estimation [58]. Although the data are multi-dimensional, quartile-based outlier removal requires scalar values. Therefore, for each sample x, we compute a scalar distance measure from the Gaussian model using the Mahalanobis distance. Quartiles Q1 and Q3 are then computed from the distribution of these distance values across all samples.”

Comment 5

With regard to the pseudocode for Algorithm 2, the feature extraction step (line 1) has now been described in the text with one added sentence, but the methodology is imprecise. Is there a paper you can cite that describes the "density-based filtering" method you've applied?

Authors Response : Thank you for the comment. We have clarified the feature extraction step in Algorithm 2 and added a citation to relevant literature describing a density-based point cloud filtering method (Deng et al., 2022) that motivates our approach. The reference is provided below.

Added : The density-based filtering step used in feature extraction is inspired by density-clustering-based algorithm proposed by Deng et al. [60] which efficiently separates ground, object, and noise like meaningful points in LiDAR data using density information .

[60] Deng, Xingsheng, Guo Tang, and Qingyang Wang. "A novel fast classification filtering algorithm for LiDAR point clouds based on small grid density clustering." Geodesy and Geodynamics 13.1 (2022): 38-49.

Comment 6

A second regarding Algorithm 2 is the definition of the "get" and "set" operations (e.g. in lines 5 and 6). It seems to me these can't simply be replacement operations (e.g. arrow meaning "becomes equal to"), as I thought that the reference frame needed to be a fusion of multiple raw frames in order to implement the equalization scheme. Bottom line, I'm a strong believer in point equalization in lidar scan matching ... but I still have no idea how this aspect of your algorithm was implemented.

Authors Response : Thank you for this helpful comment. We have clarified that in Algorithm 2 the “get” operation retrieves the current fused reference model (keyframe) accumulated from previously aligned scans, while the “set” operation performs an incremental fusion of features from the newly aligned scan into this model. These operations therefore represent access and update of a global reference representation rather than simple frame replacement.

Added : The ‘Get’ operation retrieves the current fused reference model (keyframe), which has been incrementally constructed from previously aligned scans, while the ‘Set’ operation updates this model by integrating features from the newly aligned scan.

Comment 7

In describing the results, a missing detail is a statement of the source of the ground truth. For KITTI, was GPS/INS used as ground truth to compute RMSE?

Authors Response : Thank you for pointing this out. We have added a clarification in the manuscript stating that the ground-truth poses for the KITTI dataset are provided by the benchmark and were obtained using a high-precision GPS/INS system, which we used to compute the RMSE.

Added : For the KITTI dataset, the ground truth is provided by a GPS/INS system, specifically a high-grade RTK GPS tightly coupled with an IMU (OXTS) mounted on the vehicle.

---

## [Decision Letter · Decision Letter 2]

9 Mar 2026

Density-equalized RANDT scan matching with integrated outlier removal and point density uniformity

PONE-D-25-58849R2

Dear Dr. J,

We’re pleased to inform you that your manuscript has been judged scientifically suitable for publication and will be formally accepted for publication once it meets all outstanding technical requirements.

Kind regards,

Shih-Lin Lin, Ph.D

Academic Editor

PLOS One

Additional Editor Comments (optional):

Reviewers' comments:

Reviewer's Responses to Questions

**Comments to the Author**

1. If the authors have adequately addressed your comments raised in a previous round of review and you feel that this manuscript is now acceptable for publication, you may indicate that here to bypass the “Comments to the Author” section, enter your conflict of interest statement in the “Confidential to Editor” section, and submit your "Accept" recommendation.

Reviewer #1: (No Response)

Reviewer #5: All comments have been addressed

2. Is the manuscript technically sound, and do the data support the conclusions?

Reviewer #1: (No Response)

Reviewer #5: Yes

3. Has the statistical analysis been performed appropriately and rigorously? 

Reviewer #1: (No Response)

Reviewer #5: Yes

4. Have the authors made all data underlying the findings in their manuscript fully available?

The PLOS Data policy requires authors to make all data underlying the findings described in their manuscript fully available without restriction, with rare exception (please refer to the Data Availability Statement in the manuscript PDF file). The data should be provided as part of the manuscript or its supporting information, or deposited to a public repository. For example, in addition to summary statistics, the data points behind means, medians and variance measures should be available. If there are restrictions on publicly sharing data—e.g. participant privacy or use of data from a third party—those must be specified.requires authors to make all data underlying the findings described in their manuscript fully available without restriction, with rare exception (please refer to the Data Availability Statement in the manuscript PDF file). The data should be provided as part of the manuscript or its supporting information, or deposited to a public repository. For example, in addition to summary statistics, the data points behind means, medians and variance measures should be available. If there are restrictions on publicly sharing data—e.g. participant privacy or use of data from a third party—those must be specified.requires authors to make all data underlying the findings described in their manuscript fully available without restriction, with rare exception (please refer to the Data Availability Statement in the manuscript PDF file). The data should be provided as part of the manuscript or its supporting information, or deposited to a public repository. For example, in addition to summary statistics, the data points behind means, medians and variance measures should be available. If there are restrictions on publicly sharing data—e.g. participant privacy or use of data from a third party—those must be specified.requires authors to make all data underlying the findings described in their manuscript fully available without restriction, with rare exception (please refer to the Data Availability Statement in the manuscript PDF file). The data should be provided as part of the manuscript or its supporting information, or deposited to a public repository. For example, in addition to summary statistics, the data points behind means, medians and variance measures should be available. If there are restrictions on publicly sharing data—e.g. participant privacy or use of data from a third party—those must be specified.

Reviewer #1: (No Response)

Reviewer #5: Yes

5. Is the manuscript presented in an intelligible fashion and written in standard English?

Reviewer #1: (No Response)

Reviewer #5: Yes

6. Review Comments to the Author

Reviewer #1: (No Response)

Reviewer #5: Dear authors, I am happy to inform you that all your responses are satisfactory. However, please check the visuality of all figures in the final submission. Congratulations for such noble work.

7. PLOS authors have the option to publish the peer review history of their article (what does this mean?). If published, this will include your full peer review and any attached files.). If published, this will include your full peer review and any attached files.). If published, this will include your full peer review and any attached files.). If published, this will include your full peer review and any attached files.

...

Reviewer #1: No

Reviewer #5: **Yes:** DR. TARUN KUMAR DASDR. TARUN KUMAR DASDR. TARUN KUMAR DASDR. TARUN KUMAR DAS

---

## [Editor Report · Acceptance letter]

PONE-D-25-58849R2

PLOS One

Dear Dr. J,

I'm pleased to inform you that your manuscript has been deemed suitable for publication in PLOS One. Congratulations! Your manuscript is now being handed over to our production team.

Kind regards,

on behalf of

Professor Shih-Lin Lin

Academic Editor

PLOS One